# Energy Recovery from Vinery Waste: Dust Explosion Issues

**Maria Portarapillo [1], Enrico Danzi [2,\*], Roberto Sanchirico [3], Luca Marmo [2] and Almerinda Di Benedetto [1]**

[1] Dipartimento di Ingegneria Chimica, dei Materiali e della Produzione Industriale, Università degli Studi di Napoli Federico II, Piazzale V. Tecchio 80, 80125 Napoli, Italy; maria.portarapillo@unina.it (M.P.); almerinda.dibenedetto@unina.it (A.D.B.)

[2] Dipartimento di Scienza Applicata e Tecnologia-Politecnico di Torino, C.so Duca degli Abruzzi 21, 10129 Torino, Italy; luca.marmo@polito.it

[3] Istituto di Ricerche sulla Combustione, Consiglio Nazionale delle Ricerche (CNR), Piazzale V. Tecchio 80, 80125 Napoli, Italy; r.sanchirico@irc.cnr.it

\* Correspondence: enrico.danzi@polito.it

**Featured Application: This work will help to identify and assess the dust explosion risk related to biomasses intended for energetic purposes.**

**Abstract:** The concern about global warming issues and their consequences is more relevant than ever, and the H2020 objectives promoted by the EU are oriented towards generating climate actions and sustainable development. The energy sector constitutes a difficult challenge as it plays a key role in the global warming impact. Its decarbonization is a crucial factor, and significant efforts are needed to find efficient alternatives to fossil fuels in heating/electricity generation. The biomass energy industry could have a contribution to make in the shift to renewable sources; the quest for a suitable material is basically focused on the energy amount that it stores, its availability, logistical considerations, and safety issues. This work deals with the characterization of a wine-waste dust sample, in terms of its chemical composition, fire behavior, and explosion violence. This material could be efficiently used in energy generation (via direct burning as pellets), but scarce information is present in terms of the fire and explosion hazards when it is pulverized. In the following, the material is analyzed through different techniques in order to clearly understand its ignition sensitivity and fire effects; accelerating aging treatment is also used to simulate the sample storage life and determine the ways in which this affects its flammability and likelihood of explosion.

**Keywords:** renewable energy sources; dust flammability characterization; wine waste; biomass

## 1. Introduction

### 1.1. Overview of the Role of Renewables in the Global Energy Industry

Since the industrial revolution, fossil fuels have dominated the global energy supply scene. According to IEA [1], the contribution to the total primary energy supply (TPES) of fossil fuels was equal to 81.3% in 2017, while the totality of renewable sources (considering hydro, biofuel, waste, and other minor contributions, such as solar and geothermal) reached 13%. Nevertheless, oil is still the main energy source among the others. China alone is the second energy-generating region, behind the Organization for Economic Co-operation and Development (OCDE) members put together. However, it is the leader in energy generation from coal and renewable sources (mainly hydropower, in which it is first globally) and the second region for $CO_2$ emissions into the atmosphere.

As concerns electricity generation, even though the decarbonization program has been strongly suggested and promoted by H2020 objectives in recent years, only a minor decrease in oil-generated electricity is observed (as in Figure 1). Both coal and natural gas are confirmed as the main drivers of the worldwide energy needs, and a slight growth of about 3% and 5%, respectively, among all sources from 2015 to 2017 is registered.

Nevertheless, a more significant contribution of the renewable energy supply (i.e., the sum of the contributions by biofuels, wind, hydro, solar, and tide generation) to the total electricity generation is reported (24.6%), an increase of six percentage points from 2000 to 2017. Biomass-produced electricity stands as third behind hydropower and wind power (2% of the total, 8% of renewables sources).

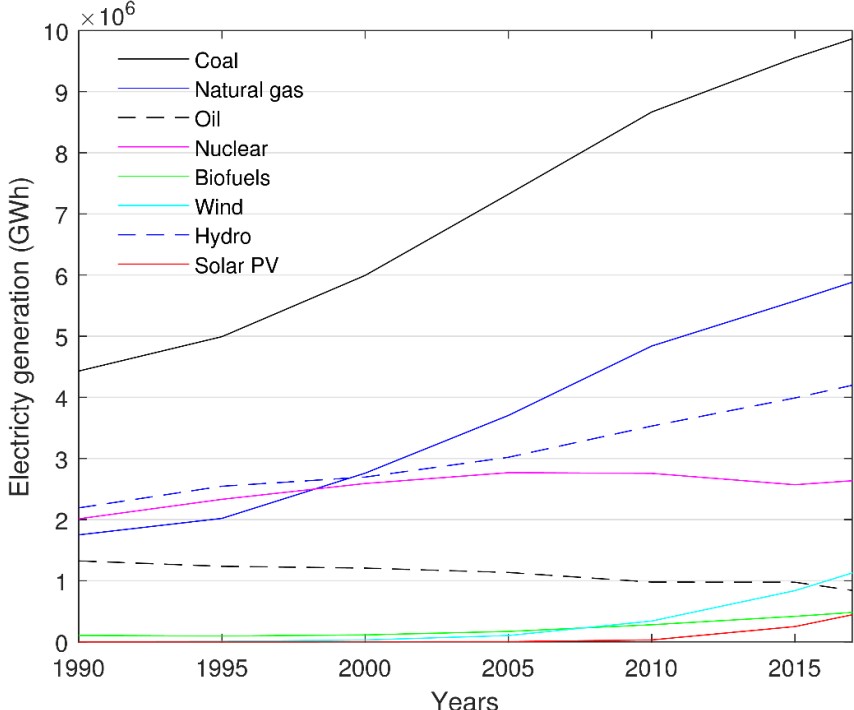

**Figure 1.** Global electricity generation by source from 1990 to 2017, redrawn from [1].

Among renewables, bioenergy is the largest source (see Figure 2). The biofuel share accounted for 1.8% in 2017, compared to the 38% and 23% for oil, coal, and natural gas, respectively, though energy generation from biofuels reported a substantial growth of about 13% between 2015 and 2017. A considerable contribution to these figures is due to the intensive use of biomass for heating and cooking in the less-developed countries (Africa has a 96% share of the total renewable supply), while an increasing trend is observed in other regions (renewables account for 59% in Europe) due to the development of biomass solutions and energetic products (such as pellets, bio-alcohols, biogases, and biodiesel) for use in electricity, transportation, and heating systems (Figure 3). Bioenergy sources dominate the heating sector: biomass in the form of wood fuel, charcoal, and residues from the agricultural sector contributed to 96% of the global renewable heat market.

A primary benefit of biomass is its easy availability in all regions: forestry and agricultural residues and/or municipal waste could be collected in every rural or urban environment. Furthermore, renewable heating from biomass is at the highest level in Europe, with an 87% share of the renewable, globally generated heating value. This value could be attributed to EU policies aiming to shift to zero-carbon sources (recently reinforced by the 2030 climate and energy framework [2]) and the prevalence of heating-district networks for residential housing. Europe is also leading the biogas and municipal waste-to-energy transformation sectors due to an intense adoption of technologies, such as gasification, pyrolysis, and direct combustion to convert biofuels into energetic materials (or directly generated power).

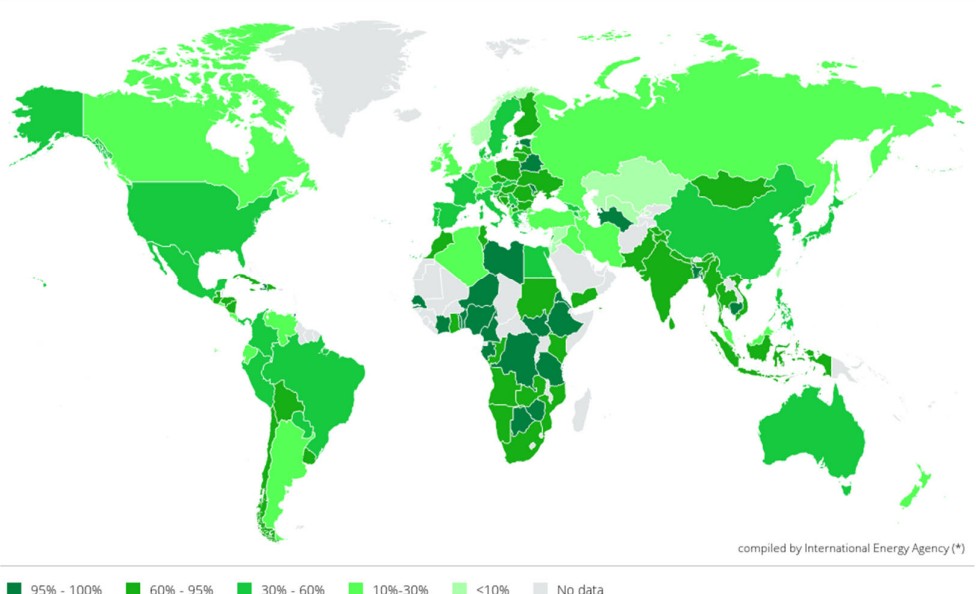

**Figure 2.** Share of solid biofuels in total renewables production (%), 2017 [1].

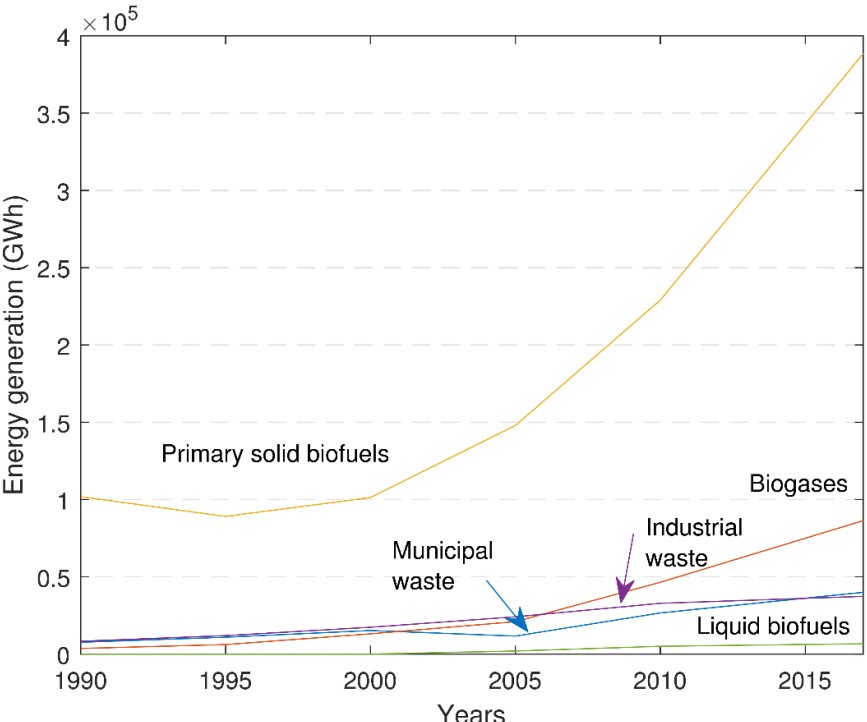

**Figure 3.** Global electricity generation by renewable source from 1990 to 2017, redrawn from [1].

Biomass supply for energy purposes could be classified according to the biomass source:

- From agriculture;
- From waste;
- From forest land.

The agricultural sector accounts for 10% of overall energy from biomasses [3], mainly related to energetic crop harvesting and biogas production from residues. Maize, rice, and wheat are the most harvested crops worldwide, in terms of area, while soybeans are the primary source of bioenergy for the Americas with respect to liquid biofuel production (biodiesel). The highest production is associated with sugarcane (1842 Mtonn in 2017). Agricultural activities are also a great source of waste, either in the form of exhaust products or as by-products, which are to be disposed of somehow, such as by energy recovering or by generating high-value secondary material. While this contribution to bioenergy is low (about 3% of the total production), an increase has to be foreseen, encouraged by the demand for fossil fuel replacement by sustainable energy sources. Municipal and industrial waste is the third source of biomass supply that could be used as fuel to feed heat systems or combined heat power ones. The biogenic fraction of waste is subjected to gasification or combustion to produce both heat and steam to generate electric power to service industrial and civil facilities. In 2017, more than half of the domestic waste supply for energy was attributed to European countries, while the US amount was a third.

The most significant contribution to bioenergy share is forest biomass used for energy purposes, including charcoal, pellets, and wood chips, accounting for 85% of the total bioenergy mix. The major contribution is associated with the pulp and paper industry and sawmill facility residues. A significant share is also related to the traditional use of wood for heating and cooking, mainly adopted in developing regions or rural areas. Furthermore, according to the WBA [3], about 11 million people are employed in the renewable-energy industry chain; bioenergy is the second biggest employer, with a total of about 29% of the total working units in 2018, including production of feedstock, transportation, conversion to biobased products for energy, equipment manufacturing, etc.

The pellets sector has recently reported a steep increase, becoming the most produced and adopted biomass commodity. More than half of the global consumption of wood pellets occurs in European countries. Among them, Italy recently ranked first for wood-pellet use for heating [4]. Its advantages include high energy density and standardized properties and ease of transport, handling, and storage. Wood and lignocellulosic material are the best candidates to generate pellets, but other promising raw materials could be used for this aim. Standard EN ISO 17225 [5] defines different materials according to their origin and source, classifying them as: woody, herbaceous, fruit, aquatic, and blend/mixture biomass. The look for alternatives in place of forest-wood fuels will reduce the impact on the forestry sector and overcome the differences in the natural biomass present in different world areas. Several industrial and academic figures focus on this direction, though the market integration of alternative biomass for pellet production, as a significant example, presents some constraints. EN ISO 17225 standard [5], while introducing the possibility of the use of wastes for pellet production, seems to be demanding in terms of product specifications and combustion properties [6].

Additionally, technical issues arise in producing energetic material from biomasses other than wood, such as the raw material's unsuitable properties (resistance to grinding or strong abrasion), difficulties in the pelletizing process operations, and lack of experience and know-how among different operators in the sector. Finally, the fire-safety issues have not been intensively considered for biomass handling and storing, apart from the solely wood industry [7], and the procedures and operations related to biomass storage, both for the energy and the food-chain industry, are as they are for the grain or flour silos [8]. Therefore, an attempt to investigate this latter point is included in this paper concerning the utilization of grape pomace as a bioenergy source. The starting points are the outcomes of a previous paper [9], where a preliminary analysis on the thermal behavior and the explosivity of grape pomace dust was performed.

*1.2. Biomass Refinery Opportunity and Limitations*

The main drivers for the further use of biomasses as energy sources (Figure 4) are multiple: the cost of the use of biomass residues to produce electricity is becoming lower than (or comparable to) that of fossil-fuel-derived technologies; their reduction effect on the GHG emissions has been proved; and the biomass supplied from agricultural, or any other human activities, could be processed as high-value material and not disposed of. Thus, the potential of a biorefinery concept is concrete: the basic idea is to utilize biomass as a platform to generate building blocks, fuels, heat, and energy. So, the several potential benefits arising from using non-renewable sources to generate renewable products are limited, and renewable biomass consumption is optimized.

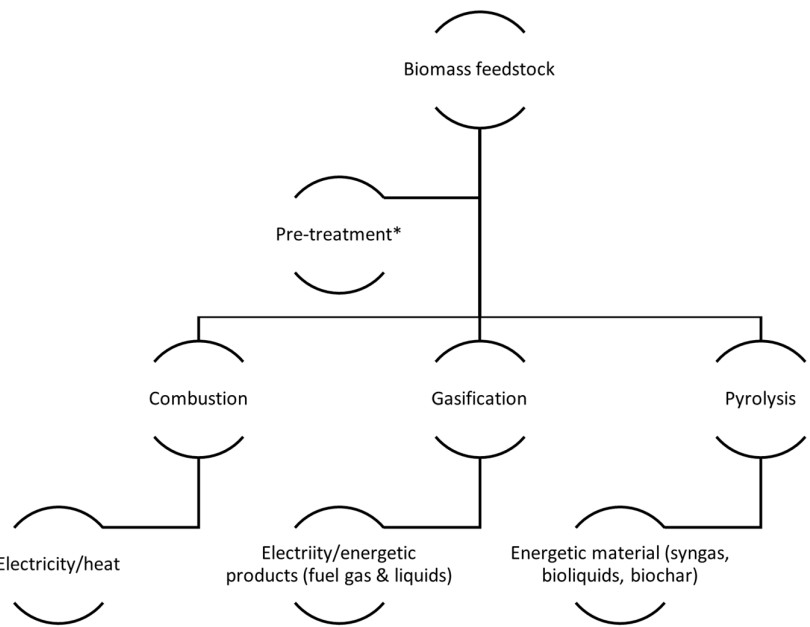

**Figure 4.** Biomass thermal conversion to energy. (*) Pretreatment depends on the raw materials, could include drying, hydrothermal treatment, torrefaction.

The cascading approach defines the opportunity to generate from biomass both low-volume, high-value products (such as phytopharmaceutical material) and many products with a limited added value (low-value, high-volume), with high energetic content, to use as fuel directly, or as fuel-carriers. Thus, the biorefinery logic focuses on the optimal use of biomass sources. At the same time, cascading implies the extensive use of biomass over time to generate, firstly, high added-value products and to recover energy from the materials only when no other processes are possible.

Both the approaches are consistent with the aim of a circular economy policy, as envisaged by the European 2030 objectives [2]. For example, the target for the reuse and recycling of municipal waste is 70%, and the landfilling maximum disposal target is 25% (of the total waste mass) in 2025.

Identifying the suitable material to be adopted as biofuel means focusing on the amount of energy stored and the form in which it is available. This energy lies in the chemical bonds of the biomass molecules, and it could be released with different transformation processes, mainly biochemical and thermal.

The main difference in adopting renewable bioenergy rather than fossil fuels is the scale of the $CO_2$ cycle in the atmosphere. While fossilized biomasses have been generated for millions of years, their recent use as fuels has released a high amount of $CO_2$ into the air: the annual rate of increase in atmospheric carbon dioxide over the past 60 years is about 100 times faster than the previous natural increases [10]. Renewable biomasses use

$CO_2$ for their metabolism and release it into the air after combustion. This cycle has a much shorter time scale than that of carbon fuels.

Key features of efficient biomass for energy purposes should be the following:

- High yield of the harvest;
- Low energy process;
- Low total costs;
- Low level of contaminants.

The total energy content is related to physicochemical characteristics of the biomass [11]. The leading indicator is the calorific value of the biomass. The Lower Heating Value (LHV) addresses the available energy content; the latent heat of vaporization is not considered as it cannot generally be recovered. Water content or moisture sensitively influences the heat content of the biomass. For this reason, thermal conversion requires a low moisture value to be effective: if biomass needs to be intensively dried before being used as a fuel, the heat balance equation is probably disadvantageous. On the other hand, biochemical processes, such as fermentation to obtain ethanol, are enhanced by the water content. The biomass' volatile matter content indicates volatile substances released during combustion (in oxygen), while the ash content is the mass remaining after burning. Both affect the likelihood of a substance being ignited and then generating heat through oxidation or gasification. Ash is the solid residue, which is not combustible. This value will affect further processing of the biomass, such as biochar production and handling.

The ultimate analysis identifies the fuel's elementary composition (C, H, N, O, and S). The ratio C:H and C:O are key factors affecting the amount of energy stored in the biomass: the higher the O and H content, the lower the amount of C-C bonds, which are more energetic (this is well represented by the Van Krevelen diagram [12], where coal and fossil fuels are in a more favorable position than biomass, due to their high number of C-C bonds). Finally, bulk density is relevant in the transforming of biomass into a fuel: denser materials are likely to be transported and stored efficiently due to their high energy–volume ratio; woody products are far more energetically dense than herbaceous biomasses, which must be compacted or packed to be used. In Table 1 the main properties of different biomasses are reported.

**Table 1.** Different biomass samples: proximate and ultimate analysis (VM = volatile matter, FC = fixed carbon, A = ash, C% = carbon percentage) and higher heating value (HHV).

| Sample | HHV [MJ/kg] | VM | FC | A | C% | Ref |
|---|---|---|---|---|---|---|
| Oak wood | 19.5 | 83.4 | 16.0 | 0.6 | 50.1 | [13] |
| Wheat straw | 17.9 | 73.0 | 17.6 | 9.4 | 45.3 | [13] |
| Rapeseeds | 19.6 | 73.5 | 19.1 | 7.4 | 47.1 | [13] |
| Olive pomace | 24.4 | 79.3 | 6.7 | 10.7 | ND | [14] |
| Grape pomace | 21.3 | 78.2 | 17.4 | 4.3 | 54.3 | [14] |
| Hazelnut shell | 19.3 | 69.3 | 28.3 | 1.4 | 52.9 | [15] |
| Tea waste | 17.1 | 85.0 | 13.6 | 1.4 | 48.6 | [15] |
| Mischantus | 19.1 | 80.3 | 16.4 | 3.3 | 47.4 | [16] |
| Spruce wood | 21.4 | 79.0 | 11.1 | 4.1 | 48.1 | [16] |
| Torrified spruce wood | 23.5 | 77.0 | 15.9 | 4.2 | 51.6 | [16] |

*1.3. Fire and Explosion Risk of the Biomass Energy Industry*

Sawdust and wood chips have been extensively used as biofuel in recent years, and industrial heating is reporting an increase in the use of this type of material as a fuel for furnaces for their heat and energy requirements [4].

Dust explosion accidents in 2019 revealed a significant percentage related to the wood and wood products industry, about 31% in the US, according to Cloney [17]. Storage

silos are significantly impacted by dust explosion, among other equipment, with a total of 13%.

Casson Moreno et al. [18] report an overview of the global risks associated with bioenergy. The data of worldwide accidents (aggregated on a three-year basis) show a similarly increasing trend in worldwide electricity generation from biomass and wastes. Most cases are related to biomass use (68%), and 48% of the episodes involved the biomass combustion facilities. An increasing trend is also observed when the data are normalized by the net electricity generation, thus suggesting that accident growth is faster than energy production. The results of their work confirm that the scale-up and the widespread use of bioenergy technology pose concerns about the associated risks, underlining the need to develop a sound and robust policy on the safety management of these types of facilities. The investigation of the explosibility of biomass dust could help raise the risk awareness and the growth of a safety culture, as is already well-established for the petrochemical energy sector.

The hazards of wood dust, as far as explosion is concerned, have been investigated in detail. Both wood materials, in the form of sawdust or finely milled, and pelletized material are currently used to fuel power plants. While the first one is intrinsically hazardous, according to several standard indications, such as in NFPA 664 [19] or ISO 80079 [20], because of the small particle size, the pellets are a granular material, which could be misguidedly considered safe, but the loading and handling operations may generate fine particles; the more fragile the original material, the finer the particles can be [6].

The IEA published a guideline on the safety assessment of the biomass industry, in which the dust explosion from wood pellet fine particles is reviewed [21].

Hedlund et al. [22] investigated an explosion case in Denmark involving a wood pellet production site, reporting on how it is essential to understand the fundamental root causes, disseminate the lessons learned, and implement safer procedures, rather than impose requirements. This approach would provide benefits to the renewable energy sector, improving its sustainability and safety.

The fundamental biomass dust explosive properties are listed in Table 2, where data derived from several dust explosion papers are listed, including our previous results on grape pomace [9].

**Table 2.** Values of deflagration index ($K_{st}$) for typical biomass dust from the literature, where F stands for Fruit, H Herbaceous, S Shells, and W woody.

| Sample | Type | d10 [um] | $K_{st}$ [bar m/s] | Reference |
|---|---|---|---|---|
| Olive pomace | F | 19 | 70 | [14] |
| Grape pomace | F | 69.6 | 57.8 | [9] |
| Barley straw | H | 25.3 * | 72 | [23] |
| Rapeseed straw | H | 31.8 * | 23 | [23] |
| Walnut shell | S | 9 | 98 | [24] |
| Pine nutshell | S | 9 | 61 | [24] |
| Southern pine | W | 28.4 | 105 | [16] |
| Northern pine | W | 25.4 | 95 | [16] |
| Pine sawdust | W | 36.5 * | 194.4 | [25] |

* this value refers to the d50.

A significant contribution to this matter comes from the research group based at Leeds University; an exhaustive document of their works is provided by the seminar meeting by Andrews G. et al. [26].

Among others, the studies by Huescar M. et al. [7,27], in which the explosion violence parameters of biomass (mainly wood-derived) are compared to coal dust ones, and a new experimental setup is proposed for the determination of the minimum explosible

concentration (MEC) of biomass powders. The same research group also investigated the influence of the torrefaction process (a standard pre-treatment technology to dry biomass feedstock before energy recovery through combustion) on the explosion severity of spruce wood [16].

Kukfisz [28] investigated the co-firing of biomass and coal in boilers. The dust explosion assessment revealed that the addition of biomass dust (in this case, straw) to traditional fuels used in CHP boilers could enhance the explosivity properties of the mixture, where both the maximum pressure values of the mixture are higher than that of the pure samples and the KSt of the mixture is higher than that of the coal sample alone.

Kopczyński et al. [29] found similar fire-propagation hazards during biomass and bituminous coal co-milling operations. The biomass was composed of sawdust wood pellets and sunflower husk pellets.

Another recent work by Pietraccini et al. [14] studied the explosibility behavior of dust samples generated by the grinding and milling of olive pomace waste.

A comprehensive study on the flammability hazards of solid biomasses is that of Garcia Torrent et al. [30], which analyses several commonly used biomass matrices with dust explosibility tests and thermal susceptibility procedures. The results are compared with those obtained with a bituminous sample (coke). While PSD is confirmed to have a key role in dust sample flammability and explosibility behavior, the oxidation characteristic Temperature ($T_{charact}$), H/C, and O/C ratios could be considered relevant for the thermal susceptibility definition.

They concluded that biomasses with a higher H to C ratio (and therefore closer to cellulose composition) but a lower O to C ratio show a greater tendency to self-ignite, in contrast with the behavior found for coals; this is explained by the authors with a different mechanism of self-heating.

The current paper aims to fill some gaps on the explosibility behavior of fruit biomass dust, in this case exhausted grape pomace, as introduced in Danzi et al. [9] and to focus on the key safety issues of dust combustion and explosion in the energy generation sector in order to optimize the risk-assessment and mitigation measures to be implemented, as was the case for metal dust in Danzi and Marmo [31].

*1.4. Waste Generation and Recovery from Grape Manufacturing*

1.4.1. Vinification Process

According to OIV [32], in 2018 the global production of wine was equal to $293.6 \times 10^6$ hl; about 19% of the total is the amount associated with Italian producers. The biggest wine industries are in Italy, France, and Spain. Their quantities of grape production and harvested areas are reported in Figure 5 [33]. China is the leading producer of grapes, while the local wine industry has not yet developed to compete globally (the most significant amount is for export).

The wine industry worldwide is one of the most relevant agricultural sectors, with a market of USD 64,100 million in 2018 and an estimated USD 72,400 million by the end of 2025 [34]. Wine has been produced since the beginning of agriculture, when mankind became settled, and it is still produced today with the same basic process. The biochemical process operated by yeasts (*Saccharomyces cerevisiae*) on the sugar content of fruits to produce alcohol is defined as alcoholic fermentation, and it consists in:

1. Scission of complex sugars into simple ones (Fructose and glucose):

   $$C_{12}H_{22}O_{11} + H_2O \rightarrow C_6H_{12}O_6 + C_6H_{12}O_6$$

2. Ethanol generation from glucose and fructose:

   $$C_6H_{12}O_6 \rightarrow 2CH_3CH_2OH + 2CO_2$$

Grape is the solid residue of fermentation; it contains only simple sugars such as fructose, and the main phase of the fermentation process is reported at point 2.

Apart from the chemical process, vinification consists of different phases, as reported in Figure 6, including mechanical operations, such as grape pressing and decantation.

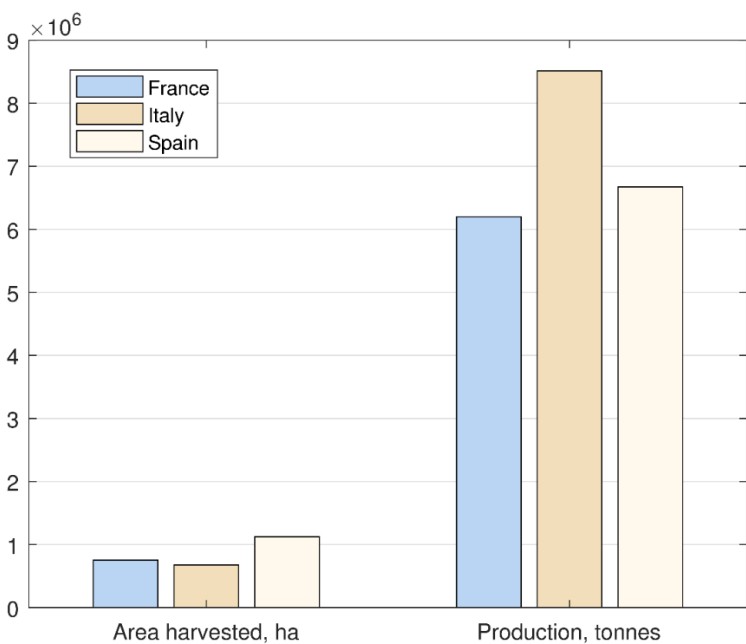

**Figure 5.** FAO [33] wine production data for Italy, France, and Spain.

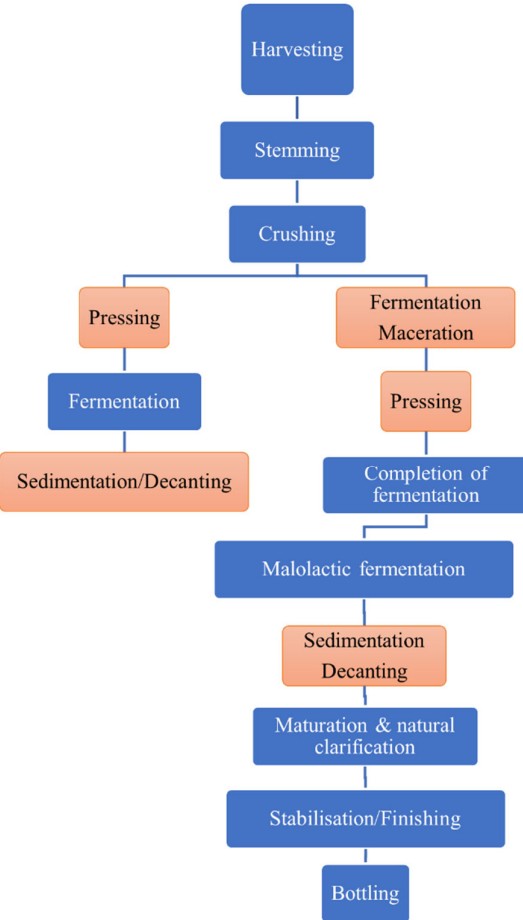

**Figure 6.** White and red wine production and waste generation. Red squares define steps, where waste flow is produced; from sedimentation/decanting operations grapevine lees are generated, while pressing produces a waste stream composed of marc and bagasse.

The vinification process is typically a seasonal operation as grapes mature in autumn and could only be collected then. First, the removal of stalks and stems is performed for white wine, and then, the residual material is conveyed to crushing machines. Crushing is done mechanically; then, the grapes are pressed to obtain must and solid residues. About 80 L of must is produced from 100 kg of grapes [35]. Then, the fermentation phase takes place, usually in stainless steel vessels, with continuous mixing to homogenize the solid parts and the liquid and adequately distribute the yeasts in the internal volume. After fermentation, the supernatant liquid (the precursor of wine) is separated from the residuals (lees) in a decanting operation and pumped to tanks where the final stabilization occurs. Wine could be clarified in the finishing phase, and then, it is ready for ageing in bottles or barrels.

In Figure 7, the different types of wastes deriving from the vinification process are described; the large size of the global wine industry produces a large amount of waste every year. Figure 7 shows the flow balance of the wine production; the percentage quantity of the solid waste produced varies depending on the references and the type of vinification (red or white), respectively, in a range between 11 to 22% and 12 to 25% of the total grape mass crushed [36]. Table 3 shows the global grape, wine, and waste production.

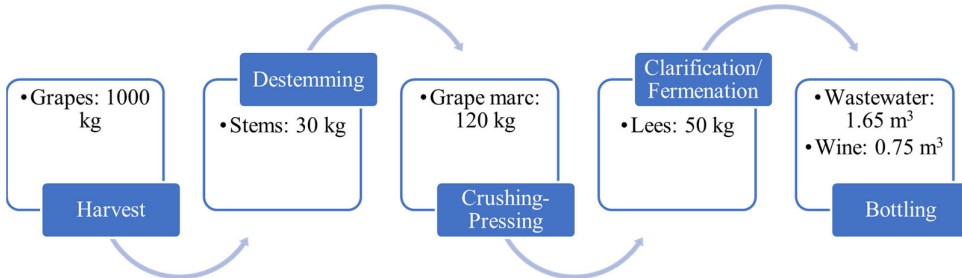

**Figure 7.** Schematic flow of vinification products, from [36].

**Table 3.** Global grape, wine, and waste production.

| Parameter | Value | Reference |
|---|---|---|
| Global fresh grape production | 75,524,194 ton | [32] |
| Global wine production | $293.6 \times 10^6$ hl | [32] |
| Solid waste per one ton of grape | 250 kg | [36] |
| Wine for one ton of grape | 0.75 m³ | [37] |
| Total solid waste, global (estimate) | 18.8 ton | - |

The solid organic waste of vinification includes stems, stalks, lees, and marc. Grape marc (or pomace) is generated after the pressing phase of the feedstock and consists of 62% of the total organic waste [36]; it is mainly composed of water, seeds, and skins, with some differences in the composition according to the vinification (Figure 8).

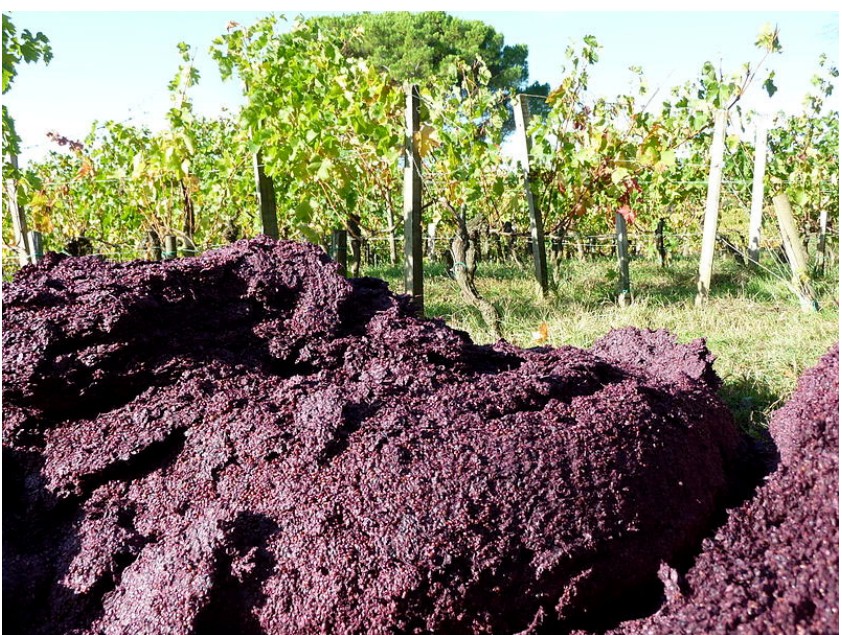

**Figure 8.** Pomace in the vineyard after pressing (Wikimedia Commons).

Regarding the pomace's general composition, the humidity varies between 50% and 72% depending on the grape variety used and the degree of ripening. As a result, insoluble residues have a lignin content of between 16.8% and 24.2% and less than a 4% protein content. In the usual way, the peptic substances are the majority polymer-type constituent of the cell walls present in grape pomace, varying from 37% to 54% of the cell wall polysaccharides. On the other hand, cellulose is the second type of polysaccharide in grape pomace, varying between 27% and 37% [38].

1.4.2. Grape Pomace Uses

According to the European Council Regulation (EC) 491/2009 [39] on the organization of the wine market, grape marc and lees must be sent to alcohol distilleries to produce exhausted grape marc and a liquid waste (vinasse) [40] (Figure 9). The size of this waste flow is relevant; the EPA [41] defines waste from vinery industries as second in importance in the food and drink sector of that which cannot be disposed into municipal landfills.

Grape pomace is traditionally reused in spirits production through distillation, generating a derived by-product, exhausted grape pomace, which has slightly different properties from the non-distilled material. However, nowadays, due to the recent decrease in alcohol sales [36], the sustainability of this process does not exist anymore, and producers could not recover the costs of the disposing of the grape pomace.

A study by Melbourne University [42] reported how grape marc is used in Australia. As of 2001, the most common technique was to dump waste or settle it in ponds (27%), send it to animal feed plants (14%), or use it for direct mulch or send it for extraction (15%). As a result, most of the material was left unutilized and disposed of as waste without any treatments.

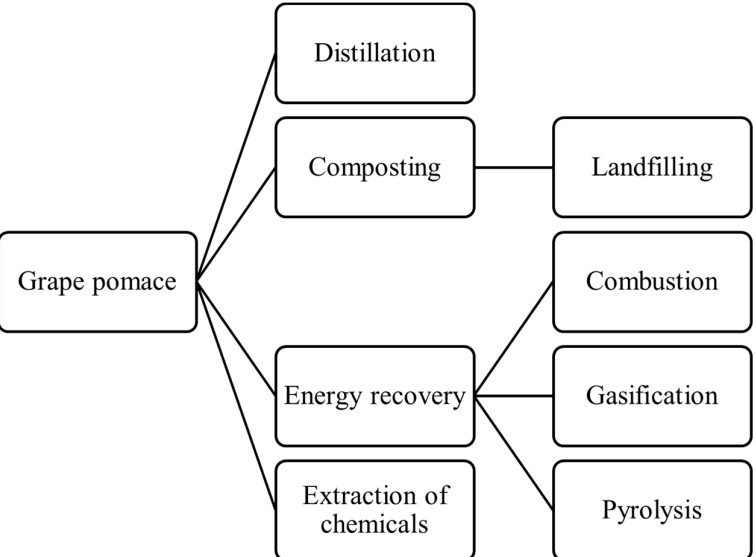

**Figure 9.** Grape pomace end-use.

Moreover, for about the 48% of the winery owners interviewed by the authors [42], little or poor concern about the environmental impact of the waste flow coming from their facilities was given; the smaller ones were not concerned about their wastes, or their perception of the problem was limited.

Even if distillation resulted in the solution proposed by the EU regulation [39], it poses similar issues: a large amount of waste is piled as storage in a short time [36], causing concerns about the generation of fermentation products, soil and surface water contamination, and potential fire hazards. Moreover, the exhausted grape pomace resulting from the distillation process still must be disposed of somehow, posing the same issues as the original material: each ton of pomace generates approximately the same amount of exhausted product.

### 1.4.3. Vinery Waste as Fuel

Combustion and thermal treatment for energy recovery are optimal, viable solutions for reusing the waste from the vinification industry. The evident advantage of biomass-generated energy is the almost total carbon neutrality, and it is reported as an interesting approach to dealing with pollution by wine waste by several studies [13,43].

Thermal conversion could be considered for the direct use of pomace inside a combustion furnace (in the form of pellets or granular bed) or to produce energetic material, such as charcoal (through pyrolysis) or bioliquids (through gasification). Furthermore, being a "second generation" fuel obtained from biomass waste, it can be converted directly into energy, without any environmental and societal issues, or economic impacts related to the allocation of dedicated energy crop harvesting.

A set of data reporting the calorific values of grape pomace could be found in the literature, and the results confirmed the potential use of it to generate energy as it has an LHV on average superior to other agricultural wastes, and similar to some fossil fuels, such as coal and coke blends (see Table 4).

A critical issue in the combustion process feasibility is represented by the high moisture content of pomace, about 50 to 60%. Thus, a share of the heat generated from the recovery process must be accounted for in the pre-treatment of the pomace as a drying or torrefaction process is to be designed; otherwise, the combustion may be ineffective. The impact on the energy production of the moisture content is estimated by [13] as being a factor of four, reducing the water content from about 70% to 10%.

Nevertheless, Burg et al. [44] analyzed the energy balance of a thermal conversion process of grape pomace, consisting of pre-drying and HTC through a steam turbine. As a result, a positive income of about 30 GWh per year could be obtained (with 30 Kton per year of material).

Grape pomace was identified among other agricultural residues as one of the most promising (after straw and corn stalks) to be converted into pellet fuel (from [45]), particularly in countries where grapes are intensively cultivated, such as in the Mediterranean areas and central and southern Europe. Moreover, grape pomace seemed to satisfy the class B pellet requirement according to ISO 17225-6:2021 [46], which allows its adoption as a fuel for medium-scale application. Furthermore, several studies ([6,47]) have demonstrated the feasibility of using grape pomace to produce pellets for heating purposes due to its suitable property values, such as LHV, mechanical ruggedness, and residual ash content.

**Table 4.** Grape residues char characterization.

| Sample Name | C | H | N | O | VM | FC | HHV [MJ/kg] | H/C | O/C | Ref. |
|---|---|---|---|---|---|---|---|---|---|---|
| Grape seed meal | 54.2 | 5.8 | 1.87 | 37.9 | 56.1 | 23.0 | NA | 1.28 | 0.52 | [30] |
| Grape pomace | 54.3 | 6.35 | 2.25 | 32.7 | 78.2 | 17.4 | 21.3 | 1.40 | 0.45 | [13] |
| Grape marc | 53.7÷54.8 *** | 5.7÷6.3 *** | 1.76÷2.59 *** | ND | 63.6÷69 *** | 25.3÷28.2 *** | NA | 1.33 | 0.52 | [48] * |
| Grape marc | 49.6 | 5.56 | 2.23 | 34.4 | 65.8 | 26.4 | 19.5 **** | 1.35 | 0.52 | [49] |
| Grape marc | 43.2 | 5.94 | 0.65 | 45.5 | 72.0 | 24.5 | 20.1 | 1.65 | 0.79 | [50] |
| Washed grape pomace | 42.9 | 9.28 | 2.05 | 38.1 ** | 67.8 | 24.7 | 19.5 | 2.60 | 0.67 | [51] |
| Dealcoholized grape marc | 51.2 | 5.53 | 2.48 | 33.6 | 65.7 | 27.3 | 20.2 **** | 1.30 | 0.49 | [52] |
| This work | 51.96 | 5.73 | 1.98 | 40.33 | 67.38 [9] | 26.98 [9] | 20.6 | 1.32 | 0.58 | |

* 4 samples; ** By difference; *** Values variable in this interval; **** LHV.

*1.5. Aim and Novelty of the Present Work*

The current paper aims to fill some gaps on the explosibility behavior of fruit biomass dust, in this case exhausted grape pomace, as introduced in Danzi et al. [9]. In the present work, we attributed the effect of the lignocellulosic components on the flammability of the sample. In particular, thanks to a complete procedure involving the thermogravimetric analysis and the analysis of produced gases, the characteristic temperatures of decomposition and the produced gaseous species were identified. Notably, some of these are flammable and responsible for homogeneous flame propagation in the event of ignition. Moreover, as the issue of biomass storage has rarely been investigated in terms of safety, the effect of aging on the sample was assessed. This effect is of crucial importance as the quantities of lignocellulosic components change over time and therefore drastically change the reactivity of the sample.

**2. Materials and Methods**

The sample analyzed in this work is a grape pomace coming from the waste stream of a wine distillery in Northern Italy. The sample is characterized by a heterogeneous morphology (Figure 10), mainly composed of aggregates, ligneous fibers, and fine particles. The size distribution is reported in Table 5.

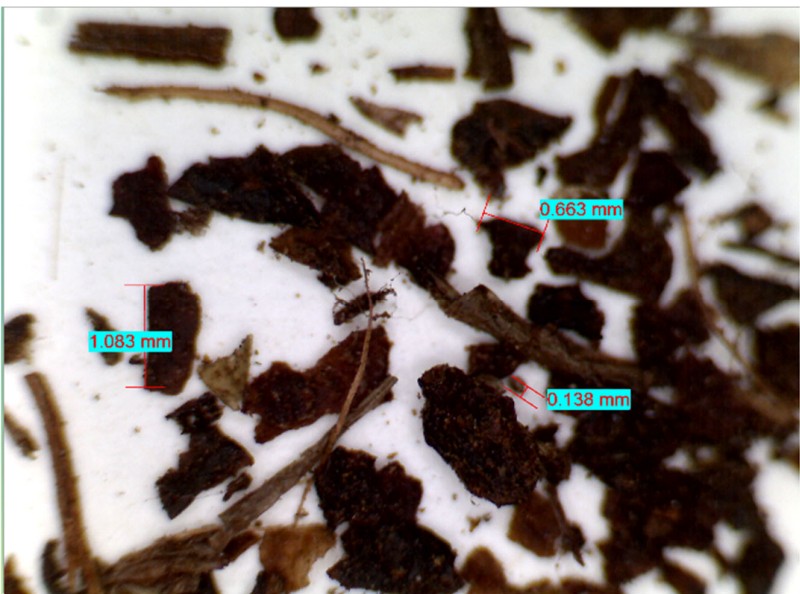

**Figure 10.** Sample morphology through optical microscopy, from [9].

**Table 5.** PSD through laser granulometry and mechanical sieving, as in [9].

| Particle Size Distribution | | | |
|---|---|---|---|
| **Laser Granulometry [um]** | | **Sieving Granulometry (wt. %)** | |
| **d10** | 69.57 | x > 2000 um | 0.04 |
| **d50** | 302.852 | 2000 um < x < 1000 um | 3.66 |
| **d90** | 633.752 | 1000 um < x < 500 um | 25.0 |
| | | x < 500 um | 71.3 |

The heterogeneity in shapes could play an essential role in the explosive behavior of the sample; it is affected by the handling and processing of the original material mixture, which behaves differently due to the different components' "resistance" to mechanical operations (grinding/crushing/milling, etc.).

The components of the original sample (grape skins, seeds, and other residues) generate particles with different shapes and are likely to have different sensitivities to ignition due to their chemical nature, as already observed for olive pomace in [14]).

The sample was originally investigated in [9], where thermal behavior and physical characterization were addressed; this work presents an in-depth analysis through alternative analytical techniques (such as TGA/DSC in an oxidative atmosphere, see Table 6), while the main focus is shifted onto two key factors:

- The lignin content of the sample.
- The effect of the application of an ageing treatment on the sample.

Both the factors have shown to be relevant in modifying the combustion behavior, ignition susceptibility, and explosive properties of biomass combustible dusts, such as in [53–58].

**Table 6.** Analytical methods used in the present work.

| Analysis | Output Parameters | Aim | Standard |
|---|---|---|---|
| Bomb calorimeter | LHV | Evaluate the potential heat power of the sample combustion | ASTM D5865-D5865M-19) [59] |

| | | | |
|---|---|---|---|
| Thermogravimetric analysis (TGA)— Inert atmosphere | DTG | Estimate the pseudo-components% (lignin, cellulose, hemicellulose) through deconvolution of DTG curve | - |
| Differential Scanning Calorimetry | MWL Temperature | Evaluate the Temperature step of different phases (pyrolysis) | - |
| Proximate analysis | VM, A, M | Compare volatiles and moisture content to other biomasses | ASTM D7582-15 [60] |
| Thermogravimetric analysis (TGA)— Oxidant atm. | IET, FET | Identify the magnitude of the exothermic reaction | - |

Starting from our previous results [9], the lignin content influence will be evaluated in the following, based on the outcomes from the thermal characterization of the sample.

Moreover, a hydrothermal ageing process will be applied to predict changes in the sample properties over long-term ageing at ambient conditions.

The aged and the as-received sample were then submitted to an analytical test to investigate whether the ageing process is effective on the physic-chemical nature of the dust and if some relevant changes appear in the aged sample, i.e., a higher susceptibility to ignition or generation of flammable gaseous products.

An FTIR was also performed to obtain early information on the sample's chemical composition and nature (Figure 11). The spectrum was compared to those of the main biomass elements: lignin, cellulose, and hemicellulose.

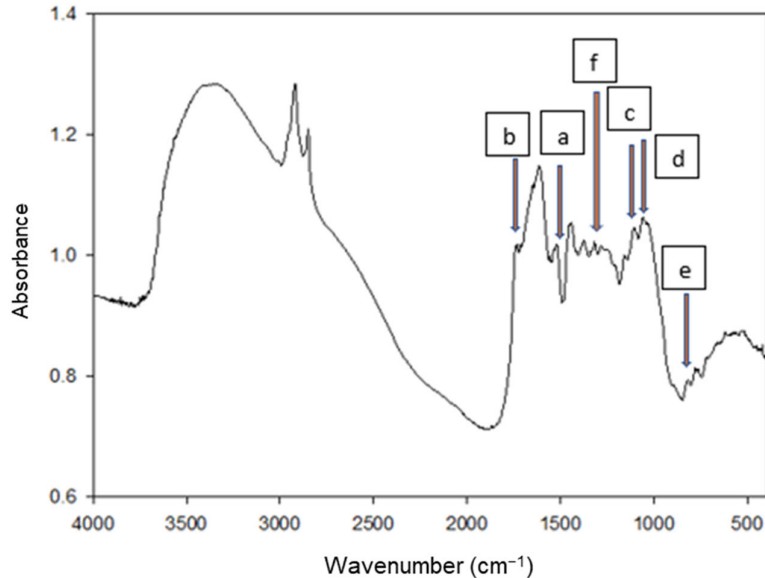

**Figure 11.** FTIR spectra of the sample, edited from [9]. Explanation for a–f callouts are reported below.

From the spectra, the following is recognizable: (a) the lignin associated C=C stretching (1510 cm$^{-1}$); (b) the hemicellulose associated C=O stretching (1740 cm$^{-1}$); (c–e) the C-O stretching and C-H deformation associated with cellulose (11,611,120, and 890 cm$^{-1}$, respectively); (f) the symmetric C-H bending associated with cellulose and

polysaccharides (1420 cm⁻¹). Similar considerations were reported by Bekiaris et al. [61] in their work on biomass substrate identification. Moreover, the typical [62] lignin fingerprint (1000–2000 cm⁻¹) is observed, qualitatively confirming the assumption reported above.

## 2.1. Thermal Characterization

Different analytical techniques were employed during this work to verify this assumption, from FTIR analysis to thermal characterization of the sample through proximate analysis, thermogravimetric analysis (TGA), and differential scanning calorimetry (DSC). Table 6 reports the tests performed and the main outputs and considerations based on them. TGA/DSC TA Instrument Q600SDT was used to perform the TGA/DSC and the proximate analysis, following the ASTM D7582-15 method [60], while as for TGA/DSC in an oxidative atmosphere, the sample was heated up to 1000 °C (heating rate: 10 °C/min) under airflow (0.1 L(STP)/min).

A deconvolution procedure of the DTG curve is used to estimate the main components of the lignocellulosic materials. The area beneath those curves represents the percentage of each component (cellulose, hemicellulose, and lignin). Fraser–Suzuki deconvolution [63] was used in this study, and the approach efficiently fit the experimental curves ($R^2$ equal to 0.99). Besides this, an ultimate analysis (according to ASTMD3176-15 [64]) was effectuated to quantify the sample's elemental composition. FTIR was used to analyze the product gases from the sample's degradation. A TGA/FT-IR interface linked by a transfer line to the TGA furnace was used for this aim. The cell and transfer line of the TGA/FTIR interface is heated and kept at 220 °C to avoid condensation of the product gases. The background is KBr, resolution 8 cm⁻¹, range 4000–400 cm⁻¹. The output of this analysis is a Gram–Schmidt diagram.

## 2.2. Hydrothermal Accelerated Ageing

Hydrothermal treatment is one method for the accelerated ageing of wood and lignocellulosic material. The ageing may reduce hygroscopicity, enhance stiffness and brittleness, and cause changes in the chemical composition [65]. Consequently, it may affect the flammable/explosible behavior of the sample.

The physical/chemical modifications due to hydrothermal treatment included both reversible and irreversible effects [66]. The reversible effects can be annulled once the sample is re-moistened. The irreversible chemical changes consist of decomposition, cross-linking, and recrystallization of the wood constituents [67]. To reproduce aged wood by hydrothermal treatments, both the effects must be considered, and the temperature and the relative humidity during heating (RHh) play a crucial role. According to Chedeville et al. [68], the chemical reactions induced by heating at 150 °C or higher are qualitatively different from those at 130 °C or lower. Therefore, 120 °C was employed as the treatment temperature because the final goal was to reproduce naturally aged wood [68].

Regarding humidity, industrial hydrothermal treatments are usually conducted either in the absence of moisture (0% RHh) or in steam (100% RHh). However, to consider both the reversible and the irreversible effects, an intermediate value of relative humidity must be used [66]. In this work, the sample was hydrothermally treated at 120 °C and RH 60% for 7 days in an autoclave [66]. The RH was calculated from the deionized water vapor pressure in the autoclave.

## 2.3. Influence of Lignin Content

Lignin constitutes the second most abundant organic polymer on earth, after cellulose. It is a three-dimensional polymer with a highly branched molecule composed of phenol units with strong intramolecular bonding. As for this, it is the basis of the

structural characterization of plant fibers; it acts as a cementing agent, making it challenging to break the bonds between vegetable cells.

Following this, it is reasonable to consider the percentage of this component in a biomass dust sample as the controlling parameter for the initiation of exothermic reactions, which initiate from the breaching of the original chemical bonds. Therefore, it is expected that ignition sensitivity parameters could relate somehow to the lignin content of the sample.

The effect is evident if the activation energy of pseudo-components is concerned; as reported by [57], it is $E_0$, lignin > $E_0$, cellulose > $E_0$, hemicellulose; the thermostability of the first derives from the strong benzene rings in the structure. Moreover, the decomposition rate for lignin is relatively slower and covers the whole pyrolytic temperature range.

Lignin content does affect the HHV of biomass, with a positive dependence; a regression equation for a non-woody sample could also be used, as reported in Demirbas [53]:

HHV = 0.0877 × (Lignin content) + 16.4951

Different biomass samples were also tested in the work of G. Torrent et al. [30]: those closer to lignin in composition (woodchips, shells) present a lower Oxidation Characteristic Temperature ($T_{charact}$), which is the temperature corresponding to a sudden loss of weight in the sample subjected to TG analysis in 100% oxygen steam. According to the authors, these samples are characterized by higher H/C and lower O/C ratios and could be considered among those more susceptible to spontaneous ignition (Figure 12).

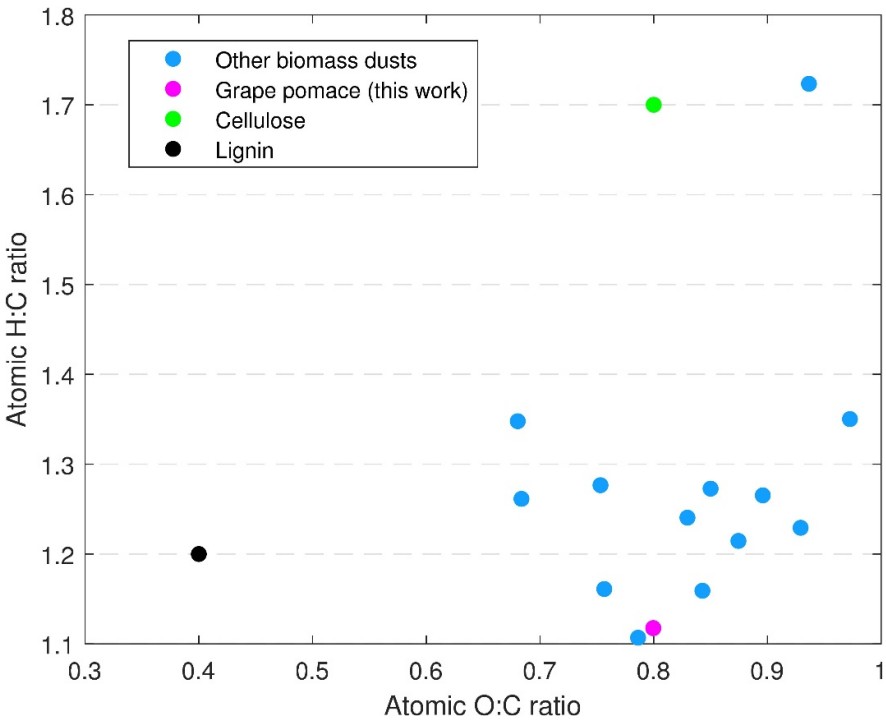

**Figure 12.** Van Krevelen modified plot for biomass fuels (from the literature [7,24,53,54]).

The volatile matter could also be linked to the composition of the sample; a low VM could indicate a low lignin content [26], and volatile species are also dependent on the pyrolysis of the components: while hemicellulose releases mainly $CO_2$, cellulose and lignin generate flammable volatiles (CO and $CH_4$, respectively) [55]. Maturity of the tree could also have a role, while lignin content tends to decrease with sample age because the youngest species tend to have a higher proportion of juvenile wood, which is richer in lignin than the mature wood [58]. However, as reported by [56], there is not a direct

connection with the lignin percentage and the MIE of the biomass dust tested in their work. Recently, Liu et al. [57] investigated the effect of lignocellulosic components and their interactions on the explosion parameters. The results showed that the explosion pressure increases with the cellulose content and decreases with the lignin one. The rate of pressure rise is mainly affected by the hemicellulose content.

### 3. Results and Discussions

In the following, the previous results from [9] (not-aged sample) and new outcomes on the aged grape pomace are presented and compared.

Figure 13 (left) shows the weight percentage and derivative thermogravimetric (DTG) curves as a function of temperature for the TG/DSC analysis in $N_2$ flow. The DTG curve exhibits two main peaks: the former at a low temperature (<200 °C) is related to the moisture loss while the latter is related to the pyrolysis of the lignocellulosic components. Following the standard test method, we carried out the proximate analysis from the weight loss curve, whose results are presented in Figure 13 (right). It is worth noting that the volatile content is very high, so it is reasonable to assume that if dispersed in air, the sample would be subject to homogeneous combustion controlled by the devolatilization process. However, it is worth noting that the peak temperature corresponding to the pyrolysis step activation is very high (about 380 °C). These data suggest that the onset of the pyrolysis could control the explosion.

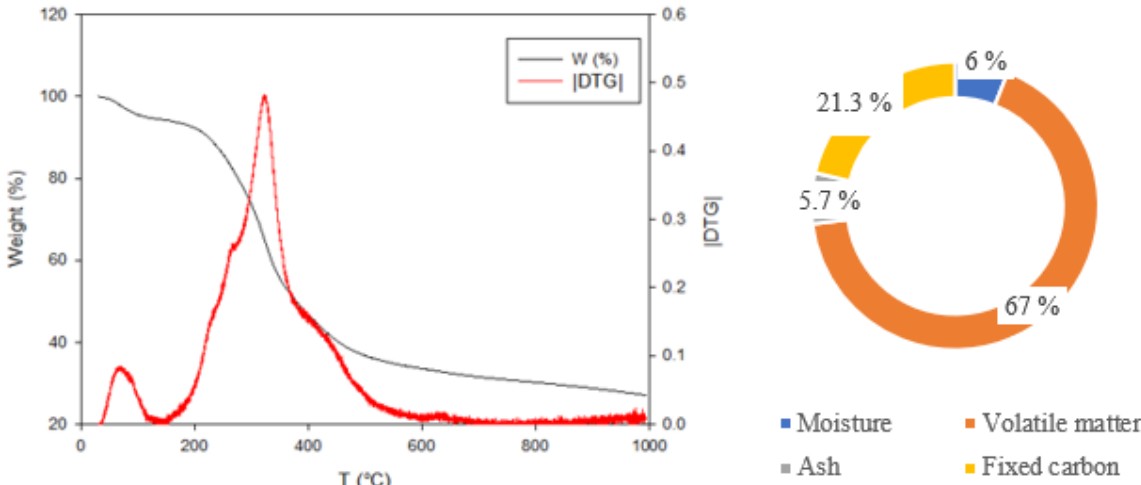

**Figure 13.** TG/DTG analysis, 10 °C/min, 100 mL/min $N_2$, 10 mg sample: weight% and DTG as a function of temperature (**left**) from [9] and proximate analysis results (**right**).

As the production of volatile species is remarkable, an FTIR gas has been carried out to evaluate which substances are released during pyrolysis. The Gram–Schmidt diagram (Figure 14) shows a single prominent peak. The time at which the peak occurs corresponds to the DTG peak temperature (320 °C). Notably, at each Gram–Schmidt diagram point corresponds an FTIR spectrum. As a result of the FTIR spectra analyses, the diagram may be divided into three zones: the first corresponding to the water desorption (FTIR spectrum not reported), the second zone is related to the decomposition, while the last section relates to the residual gas produced by the decomposition still present inside the transfer line. Figure 15 shows the FTIR spectrum at the Gram–Schmidt main peak. The thermal decomposition of the sample leads to the formation of different species, including flammable ones, such as hydrocarbon chains (from C3), carboxylic acids, and CO.

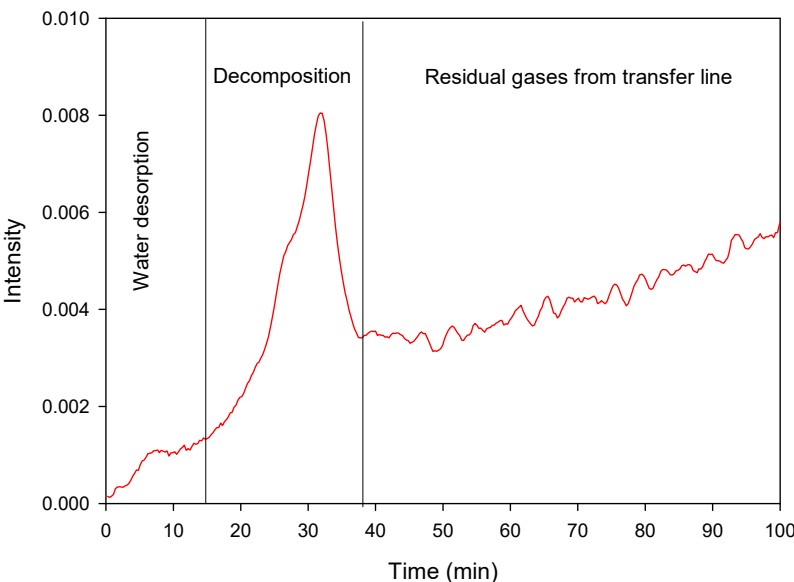

**Figure 14.** Gram–Schmidt diagram related to the TG/DTG analysis, 10 °C/min, 100 mL/min N₂, 10 mg sample.

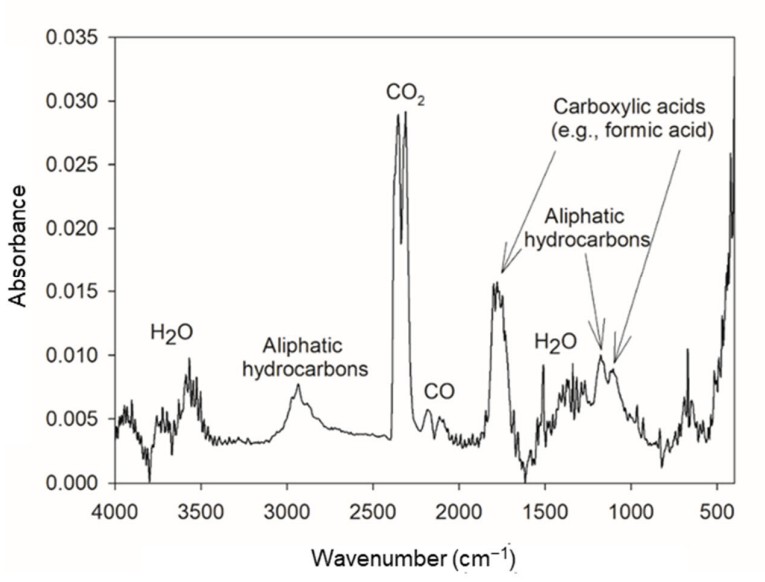

**Figure 15.** FTIR spectrum at the Gram–Schmidt diagram main peak.

Sheng and Azevedo [69] reported an analytical correlation to associate proximate and ultimate analyses values to the content in lignin and cellulose for many biomasses:

$$Cellulose = -1019.07 + 293.81 \cdot \left(\frac{O}{C}\right) - 187.639 \cdot \left(\frac{O}{C}\right)^2 + 65.1426 \cdot \left(\frac{H}{C}\right) - 19.3025 \cdot \left(\frac{H}{C}\right)^2 + 21.7448 \cdot VM - 0.132123 \cdot (VM)^2$$

$$Lignin = 612.099 + 195.366 \cdot \left(\frac{O}{C}\right) - 156.535 \cdot \left(\frac{O}{C}\right)^2 + 511.357 \cdot \left(\frac{H}{C}\right) - 177.025 \cdot \left(\frac{H}{C}\right)^2 - 24.3224 \cdot VM + 0.145306 \cdot (VM)^2$$

The analytical estimate of the pseudo-component amount was obtained from the equations above. The cellulose and lignin contents were estimated, respectively, as 5% and 61%. As observed from the FTIR spectrum, the sample mainly consists of lignin and contains a small amount of cellulose. Figure 16 (left) shows the DTG and the hemicellulose, cellulose, and lignin peaks as a temperature function, as obtained through the Fraser–Suzuki equation. The amount of each component is reported in Figure 16 (right). The results of the deconvolution are almost in agreement, as they are for lignin, with the evaluation made with the above equations. The cellulose value discrepancies could be attributed to the likely inadequacy of the method to correlate the HHV to the chemical composition (since a high variability could occur), as reported also by [69].

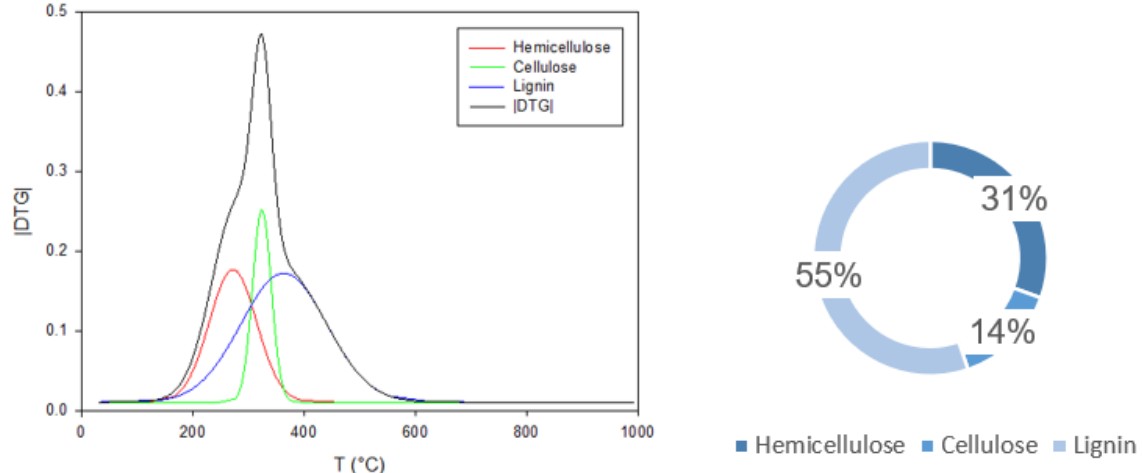

**Figure 16.** Deconvolution through Fraser–Suzuki equation of DTG curve (**left**) and lignocellulosic components amounts (**right**) of the sample, from [9].

Figure 17 shows the weight percentage and the DTG curves as a function of temperature for the TG/DSC analysis in airflow. In this case, the DTG curve exhibits three prominent peaks. The first peak is relative to the hemicellulose exothermic reaction (peak temperature 300 °C). The second is very sharp and is related to cellulose combustion (peak temperature 426 °C). The third peak relates to the lignin reaction (peak temperature 460 °C). This result makes the effect of the relative content of the three components, which have different reaction temperatures, on the flammability/explosibility parameters more understandable. The exothermic process starts at 200 °C (IET) and finishes at 525 °C (FET), with a final solid residual of 10%, attributable to char.

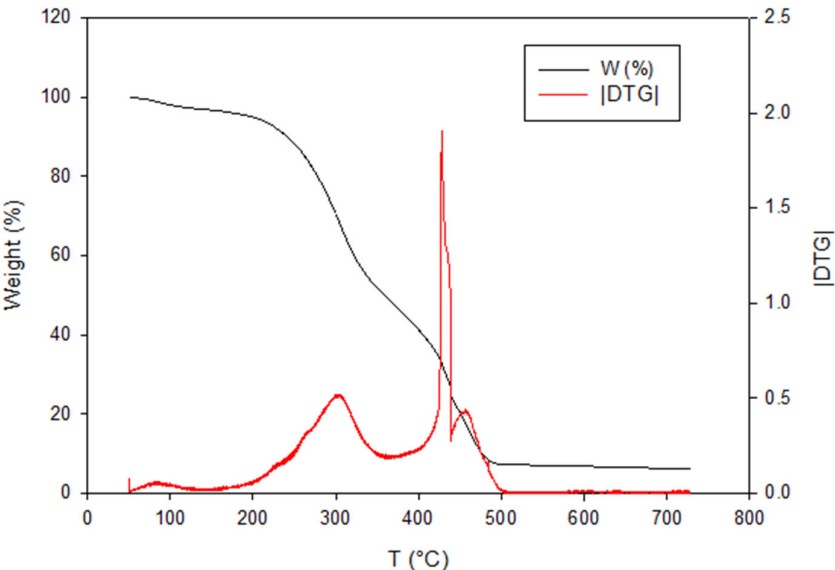

**Figure 17.** TG/DTG analysis, 10 °C/min, 100 mL/min air, 10 mg sample: weight% and DTG as a function of temperature, from [9].

FTIR gas was carried out to evaluate which substances are released during combustion. The Gram–Schmidt diagram (Figure 18) shows three peaks, as the DTG curve reported in Figure 17. The FTIR spectra (not reported) at the Gram–Schmidt peaks mainly show the peaks relative to water and $CO_2$ with CO traces in the hemicellulose combustion peak (i.e., first weight loss).

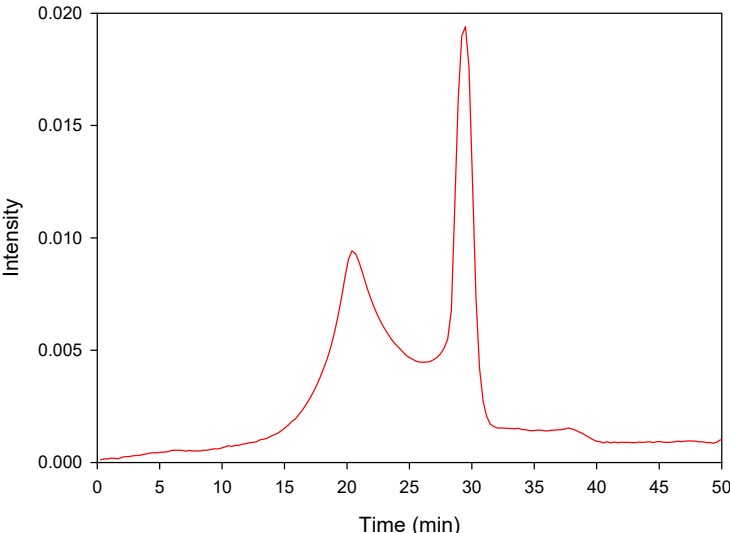

**Figure 18.** Gram–Schmidt diagram related to the TG/DTG analysis, 10 °C/min, 100 mL/min airflow, 10 mg sample.

### 3.1. Effect of Ageing

The sample was hydrothermally treated at 120 °C and RH 60% for seven days in an autoclave to assess the ageing effect on the thermal, physico-chemical, and flammable properties. Regarding the sample's appearance, analysis with an optical microscope (Figure 19) shows browning of the aged sample and a simultaneous reduction in flakes.

Grape pomace Aged Grape pomace

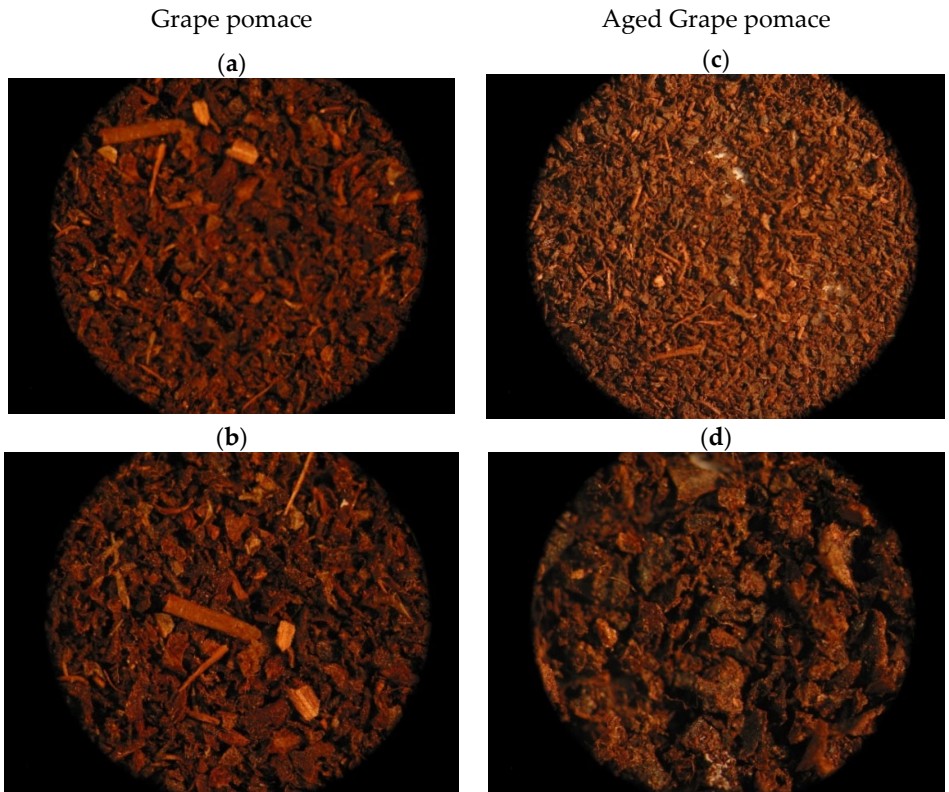

**Figure 19.** Optical microscope images with different details of the grape pomace sample (**a**,**b**) and the aged sample (**c**,**d**).

Figure 20 shows the FTIR spectra of the grape pomace sample (blue line) and the aged sample (red line), while in Table 7 the information of the bands most affected by the ageing process is reported. In the aged sample, there is a band restriction relative to hydroxyls (probably due to a water loss) and a reduction of several peaks within the wavenumber range of 1800–1200 cm$^{-1}$. As shown in Table 7, the aged sample presents a reduced intensity of the peaks related to the lignin presence (and some related to hemicellulose). The lignin and hemicellulose content decrease with the harvesting age as expected [56], reflecting an increase in the cellulose content not found from a qualitative point of view by the FTIR spectrum.

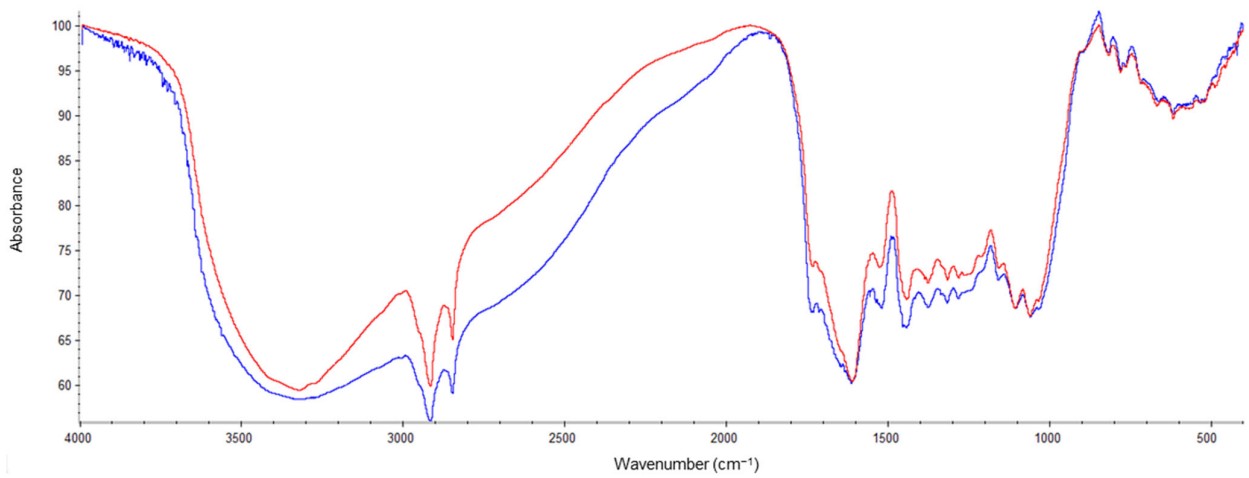

**Figure 20.** FTIR spectra of the sample (blue line) and the aged sample (red line).

**Table 7.** Wavenumbers with relative vibration and attribution affected by the ageing process.

| Wavenumber (cm$^{-1}$) | Vibration | Functional Group and/or Compound |
|---|---|---|
| Hydroxyl band | O–H stretching | Water, alcohols and phenols |
| 1740 | C=O stretching | Ketones, aldehydes, and carboxylic acids; associated with hemicellulose |
| 1650 | Absorbed O–H Conjugated C–O stretching Carbonyl C=O stretching | Water, carbohydrates, primary and secondary amides (amide I region) |
| 1510 | C=C stretching | Lignin |
| 1420 | Symmetric C–H bending O–H deformation /C–O stretching | Lignin and polysaccharides plant biomass, phenolic compounds |
| 1365 | Symmetric CH3 bending | Polysaccharides and lignin |
| 1320 | C–N stretching | Secondary amides (amide III region) |
| 1244 | C–O stretching | Hemicellulose or syringyl ring in lignin in plant biomass and wood |

Figure 21 (left) shows the weight percentage curves as a function of the temperature of the sample and the aged sample in $N_2$ flow (a) and in airflow (b). Following the standard test method, we carried out the proximate analysis from the weight loss curve, which results in Table 8. It is worth noting that the profiles are almost similar except for the moisture content within the aged sample.

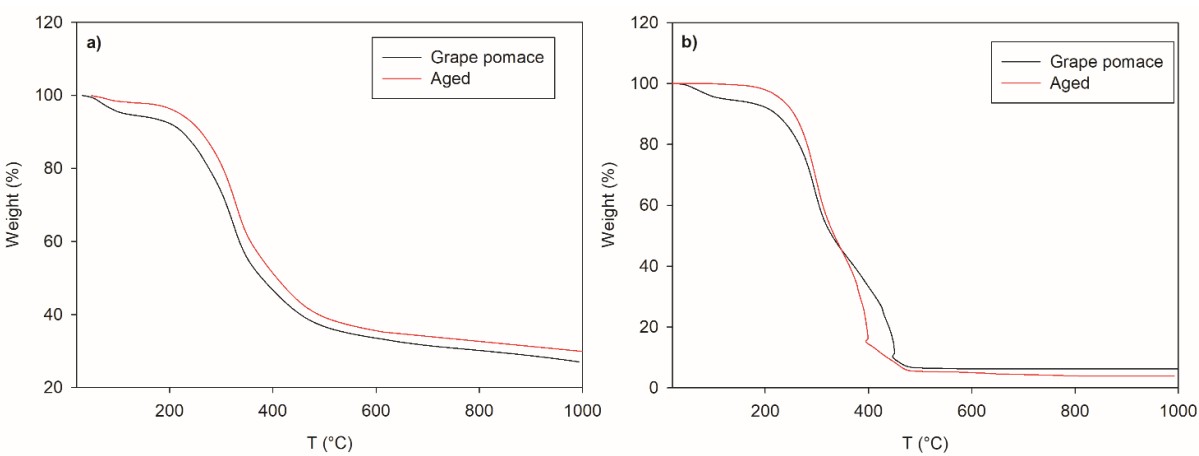

**Figure 21.** Weight% curves as a function of the temperature of the sample and the aged sample in $N_2$ flow (**a**) and in airflow (**b**).

**Table 8.** Proximate analysis.

| Samples | Moisture% | Volatile Content% | Ashes% | Fixed Carbon Content% |
|---|---|---|---|---|
| Grape pomace | 5.64 | 67.38 | 5.70 | 21.28 |
| Aged grape pomace | ≈0 | 70.10 | 5.00 | 24.90 |

Figure 22 (left) shows the DTG and the hemicellulose, cellulose, and lignin peaks as a temperature function, as obtained through the Fraser–Suzuki equation. The amount of each component is reported in Figure 22 (right). As qualitatively shown through the FTIR spectrum, ageing reduces the content of the lignin (from 55% to 48%) and the hemicellulose (from 31% to 29%) and increases the cellulose (from 14% to 23%) (Table 9).

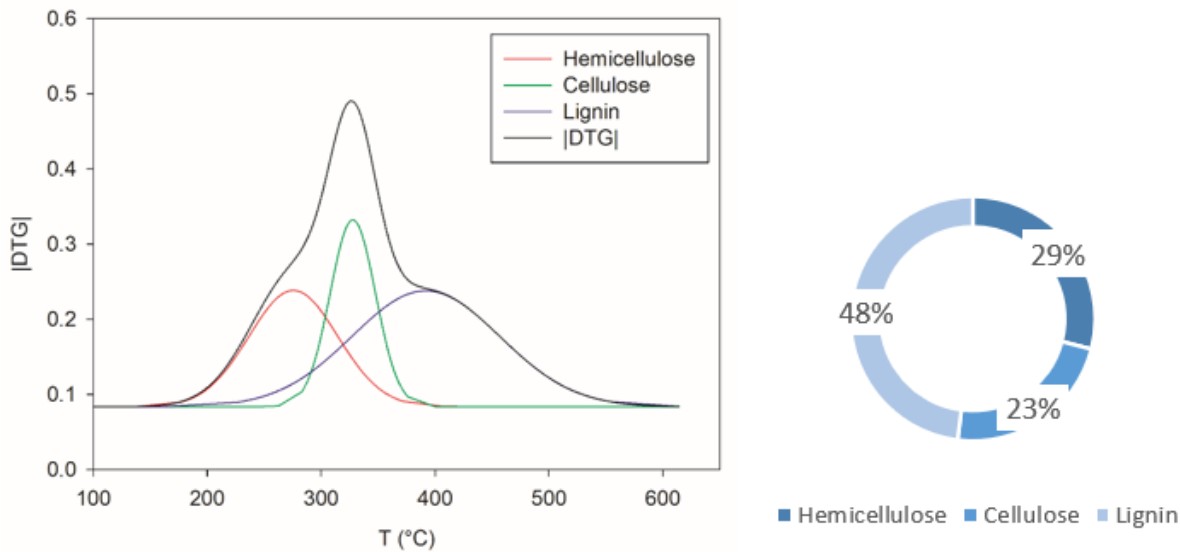

**Figure 22.** Deconvolution through Fraser–Suzuki equation of DTG curve (**left**) and lignocellulosic components amounts (**right**) of the aged sample.

**Table 9.** Result of proximate and ultimate analysis for the sample and the aged sample.

| Sample | C% | H% | N% | O% | Hemicellulose% | Cellulose% | Lignin% | H/C | O/C |
|---|---|---|---|---|---|---|---|---|---|
| As received | 51.96 | 5.73 | 1.98 | 40.33 | 31 | 14 | 55 | 1.32 | 0.58 |
| Aged | 50.45 | 5.52 | 2.73 | 41.29 | 29 | 23 | 48 | 1.31 | 0.61 |

*3.2. Explosibility Properties*

Table 10 shows the flammability/explosibility parameters of the grape pomace sample. MIE analysis showed that the sample was not sensitive to ignition by an electrical spark with an associated energy lower than 1 J. Despite the high volatile content, the volatile point is >300 °C. Regarding the MIT, no flame instantaneously came out from the apparatus, but instead, the flame appeared after 2–3 s from the injection of the powder, perhaps due to the off-gas ignition into the chamber (flash fire). Finally, from the explosibility parameters, the dust can be classified in the St-1 class.

**Table 10.** Explosibility and flammability properties of the grape pomace sample.

| Parameter | Value |
|---|---|
| Minimum ignition temperature in the cloud (MIT), °C | 480 |
| Layer ignition temperature (LIT), °C | 290 |
| Minimum ignition energy (MIE), mJ | >1000 |
| Maximum explosion pressure, bar | 6.2 |
| Deflagration index, bar m s$^{-1}$ | 57.8 |
| Lower explosivity limit, g/m$^3$ | 625 |

MIE analysis was carried out on the aged sample, and the comparison with the grape pomace sample was reported in Table 11. As can be seen, ageing leads to a reduction in the minimum ignition energy. Moreover, after the electric spark ignition, the sample shows widespread hot spots (both embers and sparks) even without flame propagation at each tested concentration. This inflammable behavior change is connected to the combined effect of the reduced moisture of the aged sample (moisture content does influence dust explosibility, i.e., the ignition sensitivity is reduced, see as main [69,70]) and the decrease in the hemicellulose content, whose decomposition generates $CO_2$. In

addition, this result may be attributable to the sample's morphological variation, which shows a lower presence of flakes.

**Table 11.** MIEs and volume-weighted mean diameters (pre- and post-dispersion) of both the samples.

| Sample | MIE (mJ) | D(4,3) (µm) | D(4,3) Post Dispersion (µm) |
|---|---|---|---|
| Grape pomace | 1000 | 208 | 324 |
| Aged grape pomace | 740 * | 211 | 254 |

* Sparks and hotspots at each concentration.

To evaluate the effect of the diameter, we used laser diffraction granulometry (Malvern Instruments Mastersizer, 2000) to characterize the granulometric distribution on both the samples, before and after the dispersion within the MIKE3. Indeed, in Table 11 the volume-weighted mean diameters (D(4,3)) are also reported. Although there is no ageing effect on the diameter, this process increases the particle cohesion (D(4,3)) after the dispersion at the walls of the MIKE3 tube after dispersion. Its morphological nature can cause the more remarkable cohesive behavior of the non-aged sample but, above all, the higher moisture content. The formation of larger agglomerates further explains the insensitivity to ignition by an electrical spark.

## 4. Conclusions

Grape pomace is a broadly available biomass, especially in southern Europe. Therefore, it represents a relevant energy source in light of the vital need to decarbonize the energy sector. Grape pomace is a granular solid with the tendency to produce fine particles. Hence, its manipulation in principle may imply an explosion hazard. This study and the previous one [9], demonstrate that an explosion hazard exists as the explosibility parameters are relevant. Grape pomace dust may belong to St1 class ($K_{st}$ about 60 bar m/s, $P_{max}$ about 6.2 bar). Ignition parameters such as MIE (>1000 mJ), MIT (480 °C) and LIT (290 °C) are typical of dust with moderate explosion hazards.

Notably, the effect of ageing is relevant. Ageing may occur in the biomass reuse processes as biomass production occurs on a seasonal basis, while its use may generally be more flattened over time. For example, aged grape pomace exhibited a lower MIE than fresh pomace (740 instead of >1000 mJ); hence, an increase in the explosion risk may be expected after ageing. In addition, the lignin and hemicellulose content decreases during ageing, which may be the principal reason for the MIE decrease. If this is the case, ageing may affect similarly other biomasses. Thus, it is crucial to consider dust age while performing an explosion risk analysis of biomasses. In prospective further studies should be the investigation of how similar biomass samples (woody, agricultural, and herbaceous among others) would behave if submitted to the aging treatment proposed here, in order to build a stronger correlation between the increase in the ignition sensitivity and the ageing of dusts.

**Author Contributions:** Conceptualization, M.P. and E.D.; methodology, M.P., E.D. and A.D.B.; validation, M.P., E.D., R.S., R.S. and A.D.B.; investigation, M.P. and E.D.; data curation, M.P. and E.D.; writing—original draft preparation, M.P. and E.D.; writing—review and editing, R.S., L.M. and A.D.B.; supervision, L.M. and A.D.B. All authors have read and agreed to the published version of the manuscript.

**Funding:** This research received no external funding.

**Institutional Review Board Statement:** Not applicable.

**Informed Consent Statement:** Not applicable.

**Acknowledgments:** The authors acknowledge Andrea Bizzarro for his excellent technical support, Ing. Massimo Urciuolo for the particle size measurements, and Ing. Giovanna Ruoppolo for the ultimate analyses.

**Conflicts of Interest:** The authors declare no conflict of interest.

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
