# Peer review of "Energy Recovery from Vinery Waste: Dust Explosion Issues"

_applsci, doi:10.3390/app112311188_

Round 1

Reviewer 1 Report

This study presents a comprehensive analysis of different aspects in relation to the potential use of grape pomace as a renewable energy source, with a focus on dust explosion risks. This manuscript could be considered as a review of the topic and can be useful for researchers and industry. However, some significant improvements are necessary before acceptance:

1. The main concerns of this reviewer are that novel information/results seem to be quite limited, since the manuscript includes several results already published by the authors (see Danzi et al, Biomass from winery waste: Evaluation of dust explosion hazards, Chemical Engineering Transactions 86, 2021) and that not all previous results are clearly identified in the manuscript. It is essential to identify and add reference in data and graphics already published, and also to emphasize the novel contributions of this paper, mainly aspects related to ageing.

2. The data provided about the effects of ageing is quite limited. Authors should indicate limitations of the research carried out so far and perspectives for the future.

3. Other comments:

- Line 35: Figure 1, although interesting, is not related to the statement where it has been placed in the main text. Some indication about the relevance of biofuels in the total renewable energy seems necessary.

- Line 44: "...slight growth of about 3 and 5%..." In which period of time? Those numbers do not seem to match information provided in Figure 2.

- Use correct notation for carbon dioxide throughout the manuscript.

- Complete information of references 13 and 28.

- Lines 269-270: use 293,6 x 106.

- Table 3: review the number for the global wine production. In addition, this number does not match the one indicated in line 269.

- Line 368: table 4 does not include LHV values, which can be relevant for wet biomasses.

- Tabla 6 and line 434 : ASTM D5142-09 or ASTM D7582-15?

- Figure 12: add reference

Reviewer 2 Report

Review (minor revision):

The article “Energy recovery from vinery waste: dust explosion issues” by Portarapillo et al. discusses the fire and explosion hazards of pulverized wine waste.

Introduction:

The introduction part describes the topics of the manuscript well. The authors show a good understanding of the subject field. Minor improvements in English are necessary. Some minor corrections are required:

Table 2: Define the KSt for the readers not familiar with explosion characteristics of dust. There is no description/definition of the * used in the table.

Figure 5: What is on the y-axis? The reference in the description is broken.

In line 269 (page 9), the number of wine produced in hl should be simplified.

In Table 3 the line, “Global wine production 2.7*108 hl” should be fixed. I assume a superscript is missing.

Figure 7: Reference.

Table 4 needs some revision. What does (53.7÷54.8) signify? The * and ** should be on the same side of the table.

Materials and Methods

  1. In line 484, what does the HHV stand for?
  2. Why is there *10 in Figure 12? Is the H:C ratio for lignin 12?

Results and discussions

  1. In Figure 14, why is the amount of residual gasses increasing? If the area from 40 to 100 min is integrated, the amount is not insignificant. Is it possible, that the signal intensity(baseline) increases due to the increasing temperature of the gasses?
  2. Formic acid (Figure 15) should present a hydroxyl peak (3500-2500 cm-1) as strong as the peak at 1750 cm-1. Why do the authors believe it is not present here?
  3. In line 546 the authors claim that the cellulose content is 5 %, from DTG deconvolution the amount predicted is 14 %. In line 551 they state that the results are in good agreement. To what do the authors attribute this discrepancy?
  4. In Figure 18 there seems to be a shoulder at approximately 37 minutes. What do the authors assume is the component responsible for this?
  5. What is the authors’ assumption for the decrease in lignin and hemicellulose content because of ageing (590-591)?
  6. In Table 8 the authors should describe what A, V, M, and FC mean.
  7. The paragraph from lines 637-642 needs to be rewritten to improve clarity. How exactly does the moisture content affect minimum ignition energy.

Conclusion:

The conclusion part is well written.

Round 2

Reviewer 1 Report

Authors have improved the manuscritp. However, several previously published results have not still been identified in this revised manuscript, such as:

- fig. 17
- left graph in fig. 13
- left graph in fig. 16
- Kst value of grape pomace in Table 2
- FTIR spectra curve in Fig. 11 (tough new interpretation for the different peaks has been added)

Therefore, my main concern about this manuscript has not been amended in a proper way by the authors. Then, my recommendation is rejection. 

Author Response

Please see the attachment,

Best regards
